# Using Deep Learning to Map Ionospheric Total Electron Content over Brazil

Andre Silva [1,*], Alison Moraes [2], Jonas Sousasantos [3], Marcos Maximo [1], Bruno Vani [4] and Clodoaldo Faria, Jr. [4]

1   Instituto Tecnológico de Aeronáutica, São José dos Campos 12228-900, SP, Brazil
2   Instituto de Aeronáutica e Espaço, São José dos Campos 12228-904, SP, Brazil
3   William B. Hanson Center for Space Sciences, The University of Texas at Dallas, Richardson, TX 75080, USA
4   Instituto Federal de Educação, Ciência e Tecnologia de São Paulo, Presidente Epitácio 19470-000, SP, Brazil
*   Correspondence: andre.silva@ga.ita.br

**Abstract:** The low-latitude ionosphere has an active behavior causing the total electron content (TEC) to vary spatially and temporally very dynamically. The solar activity and the geomagnetic field have a strong influence over the spatiotemporal distribution of TEC. These facts make it a challenge to attempt modeling the ionization response. Single frequency GNSS users are particularly vulnerable due to these ionospheric variations that cause degradation of positioning performance. Motivated by recent applications of machine learning, temporal series of TEC available in map formats were employed to build an independent TEC estimator model for low-latitude environments. A TEC dataset was applied along with geophysical indices of solar flux and magnetic activity to train a feedforward artificial neural network based on a multilayer perceptron (MLP) approach. The forecast for the next 24 h was made relying on TEC maps over the Brazilian region using data collected on the previous 5 days. The performance of this approach was evaluated and compared with real data. The accuracy of the model was evaluated taking into account seasonality, spatial coverage and dependence on solar flux and geomagnetic activity indices. The results of the analysis show that the developed model has a superior capacity describing the TEC behavior across Brazil, when compared to global ionosphere maps and the NeQuick G model. TEC predictions were applied in single point positioning. The achieved errors were 27% and 33% lower when compared to the results obtained using the NeQuick G and global ionosphere maps, respectively, showing success in estimating TEC with small recent datasets using MLP.

**Keywords:** GNSS; ionospheric models; machine learning; single point positioning; total electron content

## 1. Introduction

The users of global navigation satellite system (GNSS) single frequency applications worldwide are substantially affected by ionospheric variability because of the degradation of position accuracy caused by the phenomena in this plasma environment. This is especially true over low-latitude regions, where the dynamics are varied and intensified. Users depend on good representation of a quantity called total electron content (TEC), which is, basically, the integrated electron density along the signal path considering the volume of an abstract unitary cylinder. The TEC unit, TECu, is estimated considering the signal path as a volume of an abstract unitary cylinder. The TEC unit, TECu, is equivalent to $10^{16}$ electrons/$m^2$. Modeling the TEC is required to assess the ionospheric delay; however, over low-latitude regions that is a difficult task due to the complex electrodynamics of the ionosphere. The distribution of the plasma density over these regions varies according to several parameters such as wind patterns, hour of the day, seasonality, solar flux conditions, geomagnetic conditions, etc. [1,2]. Over low-latitude regions the phenomena include the fountain effect, the equatorial ionization anomaly (EIA), and pre-reversal enhancement of the zonal electric field (PRE), equatorial spread F, equatorial plasma bubbles (EPBs) and the

ensuing ionospheric scintillation [3]. While some of these phenomena cause mild plasma redistribution, the spread F and EPBs belong to a distinct category and correspond to sudden and drastic changes in the ionospheric F region bottom side and topside, respectively, causing GNSS users to experience the advent of ionospheric scintillation [4].

The ability to generate the TEC spatiotemporal structure properly is widely desired and would aid in providing correct estimations of the ionospheric delay. Currently, several models have been used to accomplish this task, including the Klobuchar model [5], the international reference ionosphere (IRI) [6], the NeQuick G [7], GIMs (global ionospheric maps) such as IONosphere EXchange format (IONEX) [8], neural network approaches based on long short-term memory (LSTM), multi-layer perceptron (MPL), etc., and spectral analysis techniques [9,10]. However, over low-latitude regions, especially over the Brazilian region, these models and GIMs are not capable of reproducing all the features of the ionosphere due to the large variability of the ionosphere as mentioned earlier [11]. As a result, the spatiotemporal TEC structures obtained by employing these models and GIMs offer a deficient representation. Some works explored the reasons for the deficiency of the TEC estimation comparing ground-based instruments, such as ionosondes, and TEC obtained from GNSS observables. Belehaki, A. [12] used digisonde data and GNSS measurements to estimate the contribution of the plasmaspheric density to the TEC values over Europe, where, typically, there are no EIA, EPBs, etc., and the behavior of the ionosphere changes more smoothly with space and time. The environment over the Brazilian region, however, is considerably distinct, and data from GNSS would be preferable because it includes the plasmaspheric contribution.

Nowadays, the usage of machine learning techniques attempting to predict the behavior of the ionospheric environment through TEC has been a promising approach according to recent works in the literature. Ghaffari Razin, M. R.; Voosoghi, B. [13] developed a method of ionospheric tomography based on wavelet neural networks, which showed success in estimating TEC when compared to test stations in Iran. Mallika et al. [14] presented an ionospheric forecasting algorithm combining principal components analysis (PCA) and neural networks to forecast ionospheric TEC values in a region over Japan using 20 years of TEC data along with geomagnetic activity index Ap and the solar flux index (F10.7). Liu, L. [15] used a LSTM (long short term memory) neural network to perform the forecast of the spherical harmonic coefficients applied in the construction of global ionospheric maps (GIM). Similarly, Ref. [16] presented results predicting the TEC on a global scale 24 h prior to the real data. Their work was based on GIM maps that fed a nonlinear autoregressive exogenous neural network with external input (NARX). This model was considered satisfactory, causing errors between 3 and 5 TEC units. Han, Y. [17] applied four different types of machine learning models to forecast ionospheric TEC using three international GNSS service (IGS) monitoring stations located in low-latitude regions. They evaluated the performance of these models in geomagnetic disturbed conditions with high solar activities and showed that in these scenarios it is difficult to properly predict the TEC values.

As previously mentioned, the Brazilian region is in a sector with several particularities in the ionosphere, which have been the subject for several recent research works [18]. Motivated by this complex ionospheric environment, in this work, the neural networks procedure applied was designed aiming to provide a TEC prediction that preserves essential features of the TEC spatiotemporal structure over this sector. To do so, real maps of TEC from the region of interest in periods under similar conditions were used to feed the neural network. In addition, the approach proposed has the advantage of allowing the organization and distribution of TEC forecasts to users promptly. The contributions of this work are to use deep learning to predict a more detailed behavior of the TEC over a large territorial extension like Brazil, and to provide immediate short forecasts. These features have potential to make this prediction a service of interest for users of satellite positioning technologies at low latitudes.

## 2. Methodology

The parameter to be estimated by the model proposed in this work is the TEC across the Brazilian territory. To make this prediction using a machine learning approach, real TEC values from Rede Brasileira de Monitoramento Contínuo dos Sistemas GNSS (RBMC), a network of geodetic receivers over Brazil, were used to build TEC maps over the region of interest [19]. These TEC maps were built based on the methodology of [20]. In addition to the TEC maps used as inputs to the neural networks, other parameters of space weather were also used to optimize the neural network: the solar flux F10.7 and the geomagnetic index Ap. The F10.7 index is an indicator of solar flux and measures the noise generated by the sun in the wavelength of 10.7 cm in solar flux units (s.f.u.) being 1 s.f.u. = $10^{-22}$ $Wm^{-2}Hz^{-1}$. The Ap index provides the level of geomagnetic activity over the globe and its values are obtained by averaging 8 samples during the day.

### 2.1. TEC Data Processing

One of the key factors for an accurate GNSS positioning is a fair estimate of the ionospheric delay. The ionospheric delay, however, is directly related to the TEC above-mentioned. The TEC is responsible for causing changes in the code and carrier phase measurements; this relation may be defined as:

$$I = \frac{K}{2\,f^2}\text{TEC} \tag{1}$$

where $I$ is the ionospheric delay in meters, $K$ is a constant equal to 80.62 $\left(m^3/s^2\right)$ and $f$ is signal frequency in MHz. This relationship between the TEC values and the ionospheric delays is invertible, i.e., it is also possible to obtain an estimate of the TEC values from the ionospheric delay values. To estimate the TEC, the use of dual-frequency GNSS receivers is needed. From the pseudorange and carrier-phase measurements provided by these receives, TEC may be estimated using the observables provided, typically, in the L1 and L2 frequencies (nowadays L5 may also be used). The pseudorange equation is given by [21]:

$$P_k = \rho + c\,(\Delta t_r - \Delta t_s) + I_k + T + b_{kr} + b_{ks} + m_k + \varepsilon \tag{2}$$

where $P_k$ is the pseudorange observation corresponding to the frequency reference $k$, $\rho$ is the geometric distance from the receiver up to the satellite, $c$ is the speed of the light at the vacuum, $\Delta t_r$ and $\Delta t_s$ are the receiver and satellite clock errors, $I_k$ is the ionospheric delay, $T$ is the tropospheric delay, $b_{kr}$ and $b_{ks}$ are the instrumental biases for the receiver and the satellite, respectively, $m_k$ is the term associated with multipath effects, and $\varepsilon$ is the thermal noise.

The TEC that may be obtained with this procedure is often referred to as slant TEC (STEC). To obtain the STEC from the combination of the frequencies L1 and L2, the difference between P2 and P1 is calculated, where P1 and P2 are the pseudoranges for the L1 and L2 carriers, respectively. By neglecting the multipath and thermal noise, the other terms in Equation (2) are canceled. After this, by replacing the terms related to the ionospheric delay (I1 and I2) it is possible to obtain the equation of STEC from pseudorange measurements using $f_1$ and $f_2$ [22]:

$$STEC_{code} = 2\,\frac{(f_1\,f_2)^2}{K\,(f_1^2 - f_2^2)}\,(P_2 - P_1) - b_r - b_s, \tag{3}$$

where $STEC_{code}$ represents the STEC value measured from pseudorange code observations.

To carrier-phase, the equation is given by [21]:

$$\phi_k = \rho + c(\Delta t_r - \Delta t_s) - I_k + T + b_{kr} + b_{ks} + \lambda_k\,N_k + m_k + \varepsilon_k \tag{4}$$

where $\phi_k$, is the carrier-phase observation corresponding to the frequency $k$, $\lambda_k$ is the respective wavelength and $N_k$ corresponds to the cycle ambiguities. To obtain STEC from carrier-phase, the difference between $\phi_1$ and $\phi_2$ (the carrier-phases for L1 and L2, respectively) is used. The process is like that for pseudoranges, assuming $\phi_k = L_k \lambda_k$. The equation to STEC from carrier-phase measurements using $f_1$ and $f_2$ is given by [22]:

$$STEC_\phi = \frac{2 (f_1 f_2)^2}{K (f_1^2 - f_2^2)} (L_1 \lambda_1 - L_2 \lambda_2) - b_r - b_s - (\lambda_1 N_1 - \lambda_2 N_2) \tag{5}$$

where $STEC_\phi$ is the STEC obtained from phase-code measurements.

For this kind of application, the ionosphere is assumed as a thin shell located at a reference altitude of 350 km. This altitude is used because it is approximately where the electron density peak is located and is usually referred to as the ionospheric pierce point (IPP). The models used for ionospheric delay correction in single-frequency GNSS receivers, for example, use this assumption. In this representation, the STEC is converted to vertical TEC (VTEC), because it maps the ionosphere in geographic coordinates, also called sub-ionospheric points. This conversion is given by the following equation [23]:

$$\frac{STEC}{VTEC} = \left[ 1 - \left( \frac{R \cos E}{R + h} \right)^2 \right]^{-1/2} \tag{6}$$

where $R$ is the Earth radius (in km), $h$ is the height of the thin shell representing the ionosphere (in km), and $E$ is the elevation angle of the satellite.

This work employs TEC maps produced using this approach to feed a neural network which has the purpose of predicting the TEC spatiotemporal structure for days in the near future. The input TEC map was built using data from the RMBC network over the Brazilian territory. This network currently has more than 100 receivers operating and the RINEX (receiver independent exchange) data recorded by these receivers are available at the RBMC website for download.

These maps were made according to the methodology developed by [20]. The construction of the TEC maps followed the steps given below:

1. Conversion of STEC to VTEC considering only data collected by satellites with elevation angles above $20°$.
2. All the IPPs from the VTEC obtained at all available stations are gathered during 5-min intervals.
3. At each 5-min, the IPP points are grouped into grid cells in a mesh with $1°$ resolution for the longitude $\times$ latitude plane at the IPP altitude.
4. For each grid cell, the average VTEC value is weighted by the elevation angle.
5. The Delaunay triangulation [24] process is applied using linear interpolation over the covered area. This interpolation is intended to fill regions with empty grid cells.
6. In the last step, a Gaussian low-pass filter is applied to the domain to smooth the grid transitions in the TEC map.

After executing all these steps, considering the 5-min intervals, it is possible to obtain 288 maps for a day. These TEC maps are capable of exhibiting all the macroscopic variations expected over the low-latitude region.

Figure 1 shows 6 panels with an example of these TEC maps that will be used as inputs to the neural network. The upper/lower panels present the TEC maps for winter/summer solstice conditions, hence, covering varied seasonal conditions. The left/middle/right panels correspond to morning/afternoon/early nighttime periods, respectively. The upper panels describe smaller TEC concentration and closer to the geomagnetic equator, as expected for a winter solstice period. The lower panels display a different behavior with larger TEC values and plasma being redistributed to low latitudes, away from the equator by the fountain effect, thereby forming the equatorial ionization anomaly at afternoon and early nighttime.

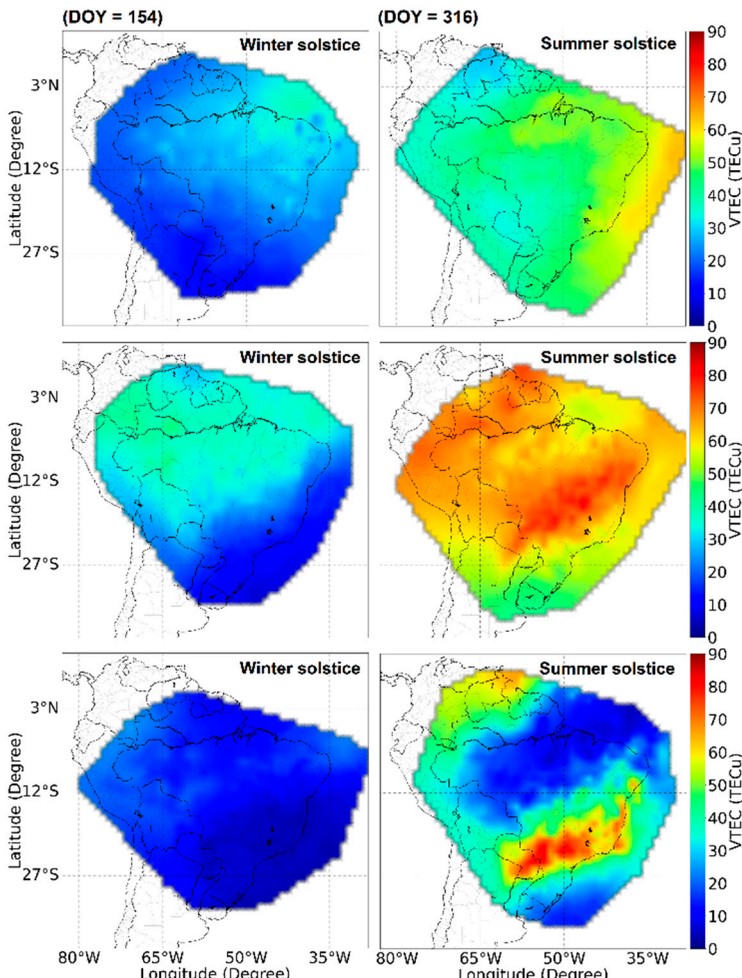

**Figure 1.** Example of TEC maps used as input to the neural network to be discussed in the later sections. **Right/left panels**—winter/summer solstice example (3 June and 20 November, respectively). **Upper/middle/lower panels**—TEC map at 13h00UT/20h00UT/23h00UT (morning/afternoon/early nighttime for the Brazilian region).

## 2.2. Reference Data and Metrics

The performance of the ionospheric delay estimation from the TEC prediction from this work was compared with the NeQuick G model and the GIM provided by IGS. NeQuick G is the model used by Galileo (European navigation system) for ionospheric delay correction. The NeQuick G model calculates the correction using the daily solar flux information transmitted through 3 coefficients obtained from the navigation message [24]. The GIM IONEX is a product generated by the IGS and contains global VTEC information and DCB measurements from GPS satellites. The final GIM product has a latency of approximately 11 days, and the files have a temporal resolution of 2 h with a spatial resolution of 2.5° in latitude and 5° in longitude [25].

The results (errors) from the neural network model trained with the real TEC maps built according to [20] were compared with the NeQuick G model and GIM employing statistical evaluation. For comparison purposes the adopted metric is the MAE (mean absolute error) [26]. The MAE may be described by the following expression:

$$MAE = \frac{1}{N} \sum_{j=1}^{N} \left| \vartheta_j - \varsigma_j \right| \tag{7}$$

where $\vartheta_j$ and $\varsigma_j$ are the true and predicted values of the sample $j$ in the vector of length $N$, respectively. All the analysis to be presented use error metrics based on the MAE.

## 3. Neural Network

Deep neural networks are tools able to learn from big datasets and generate models capable of mapping inputs to outputs even when there is nonlinearity in this mapping. Due to this capacity of extracting features of a dataset, this deep neural network can be used for data forecasting, for example, temperature, stock price, etc. [27]. In the case of this work, they were used for forecasting the ionospheric distribution.

Problems that involve forecasting are dependent on time; therefore, they are more complex than the classical problems in classification and regression. The multilayer perceptron (MLP) class was chosen to train a model to predict the daily maps of the TEC distribution over the Brazilian region, e.g., it predicted the VTEC values along the Brazilian region. This type of neural network architecture was chosen due to features like robustness to noise and nonlinearities, multivariate inputs, besides being simpler to implement [27].

### 3.1. Neural Network Architecture

The architecture multilayer perceptron (MLP) was applied in this work to predict TEC values. This type of neural network is composed of layers containing neurons. The layers are connected in sequence and the output of each neuron of a layer is connected to the input of all the neurons in the next layer (fully connected). The layers located between the first layer and last layer are called hidden layers. As mentioned earlier, this type of network is robust to input noise and it is capable of modeling both linear and non-linear problems [28]; therefore, the behavior of the spatiotemporal distribution of the TEC may be properly assessed.

The neuron model applied in the MLP is given by:

$$y = f\left(\sum_{i=1}^{N} x_i \, w_i + b\right) \tag{8}$$

where: $f(\ )$ represents an activation function, which relates the inputs with the output $y$; $x_i$ are the inputs of each neuron $i$; $w_i$ represents the weight of each neuron; and $b$ denotes the bias [29].

Once the neural network is assembled the training is composed of two stages: forward phase and backward phase. In the forward phase, the weights are fixed, and the input data propagate through the network until reaching the output producing an error, which is generated by comparing the output data with the desired response. In the backward phase, this error is used to adjust the network weights; then it is propagated in the backward direction. The adjustments are made successively from the last layer to the first layer at various times until reaching the desired error threshold in the output, when the training of the network is completed [29].

### 3.2. Neural Network Configuration

In this section, the configuration and parameters used in the MLP neural network used in this work are presented. The neural network (MLP) implemented has five layers and the configuration of each one is shown in Table 1.

The configuration presented in Table 1 shows the number of neurons and the activation function applied in each layer was the rectified linear unit function (ReLu). ReLu was used because this function works only with positive values and rectifies negative values to zero. Because TEC is density and always has positive values, this function returns adequate outputs.

**Table 1.** MLP configuration.

| Layer | Number of Neurons | Activation Function |
|-------|-------------------|---------------------|
| **1st** | 500 | ReLu |
| **2nd** | 100 | ReLu |
| **3rd** | 100 | ReLu |
| **4th** | 50 | ReLu |
| **5th** | 1 | ReLu |

The inputs were arranged according to the following parameters: year; day of year (DoY); time (in seconds); latitude of the IPP (in degrees); longitude of the IPP (in degrees); F10.7 index (in s.f.u.); and Ap index (in nT). All these input variables were normalized by the respective maximum value. The output variable is the vertical TEC value (VTEC). Figure 2 shows the structure of the multilayer perceptron neural network (MLP-NN) used in this work.

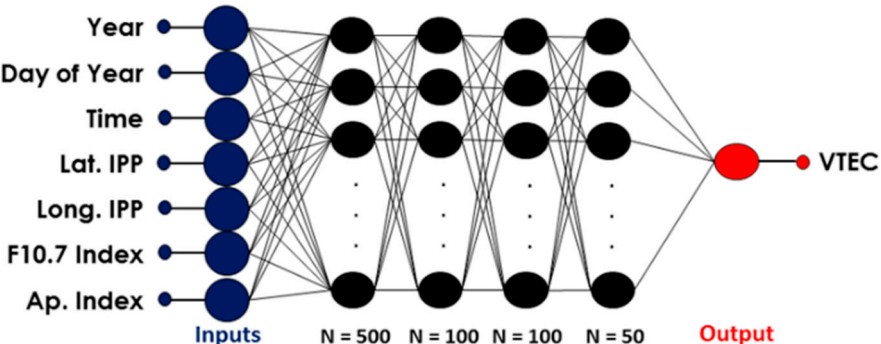

**Figure 2.** Multilayer perceptron neural network (MLP-NN) architecture for TEC prediction across Brazil.

The deep learning-based TEC model was developed using the TensorFlow framework. The neural network presented in Figure 2 was configured to avoid overfitting using the EarlyStopping function to monitor the metric MAE during the training. If the MAE achieves a value smaller than $5 \times 10^{-5}$ TEC units (TECu), the training process is interrupted before completing all epochs set in its beginning. The optimizer used to train the neural network was the Adam optimizer. This algorithm has the function of updating all neural network weights interactively using the training data. The hyperparameters for the Adam optimizer were a learning rate of 0.001, $\beta 1 = 0.9$ and $\beta 2 = 0.999$. $\beta 1$ and $\beta 2$ are, respectively, the exponential decay rate for the 1st and 2nd moment estimates of the stochastic gradient descent method of optimization from the Adam class. To compile the model of the neural network, the cost function used was the MSE (mean squared error), and the validation metrics used were MAE, MSE, and accuracy. To fit the model, a value of a batch size of 256 and a number of epochs equal to 20 were used. The training process took approximately 40 min on average using Google Collab. In addition, during the training process, whenever a given MSE achieved a value smaller than that from an MSE from a previous epoch, the neural network weights were saved, so that the best model was obtained at the end of the training process.

### 3.3. Preprocessing and Training Methodology

The training methodology was developed to predict the ionospheric VTEC of the Brazilian region; due to this, the MLP-NN training used VTEC maps that were processed using the methodology developed by [13]. This methodology permits obtaining the VTEC maps from the observational RINEX recorded from all dual-frequency global positioning system (GPS) receivers maintained by the RBMC.

The TEC maps were originally sampled every 5 min for each day available. Following, these maps were structured in tables where the columns are divided as follows: 1—Year; 2—DoY (day of year); 3—time (seconds); 4—latitude IPP (degrees); 5—longitude IPP (degrees). Then, the F10.7 (10–22 W m$^{-2}$ Hz$^{-1}$) and the Ap (nT) were concatenated (per day) in columns 6 and 7, respectively. These space weather indexes were obtained from the OMINIWeb website (https://omniweb.gsfc.nasa.gov/form/dx1.html, accessed on 14 December 2022) maintained by the National Aeronautics and Space Administration (NASA). The VTEC from each day was separated in a single column to compute the error during the training stage.

The training strategy was based on datasets of five days prior to the given day that should be predicted; by doing so the dataset was fully used sequentially, i.e., each day predicted was the result of the training using over the five days prior to this desired day. This procedure was performed for the entire year of 2014. Please observe that the aforementioned approach implies outputs from January 6 up to December 31. At the end of this process, the values were denormalized and the TEC values were obtained from the neural network.

To exemplify the performance of the neural network during the training phase, Figure 3 presents the cost function and the MAE as functions of time during the training of the VTEC prediction for day 365 of year 2014.

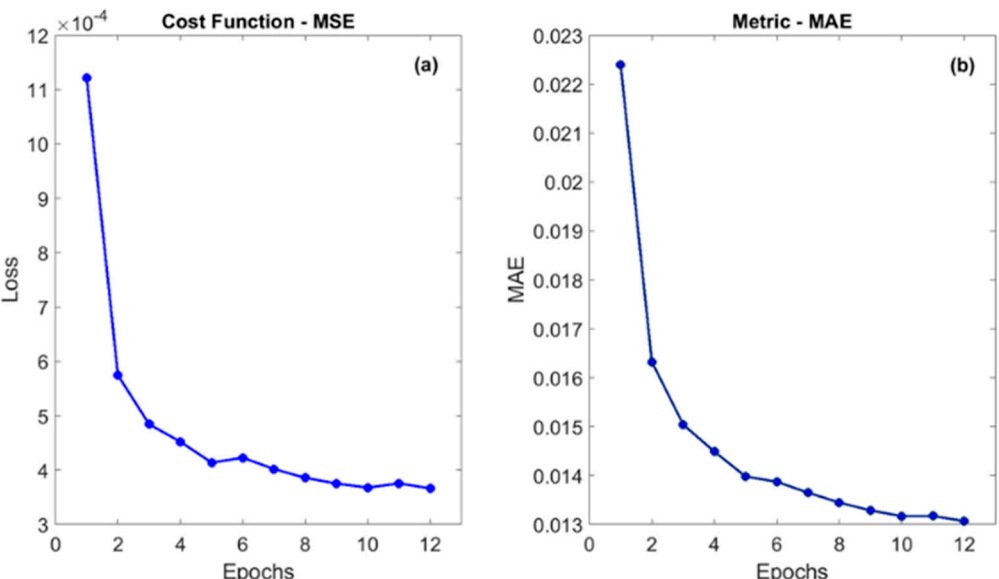

**Figure 3.** Panel (**a**)— cost function (MSE) for day 365 of year 2014. Panel (**b**)— metric (MAE) for the same day. According to the panels the convergence was reached after few epochs of training.

Panel (a) of Figure 3 presents the convergence of the cost function (MSE). Please note that the value of the cost function drops quickly as the epochs advance. Panel (b) shows the convergence of the MAE, used later to evaluate the error in the estimation of the VTEC value (normalized). The MAE value also drops considerably according to the advance of the epochs up to epoch 12, where the EarlyStopping function stops the training. Please remember that the function was configured to stop the training when the MAE value was less than $5 \times 10^{-5}$. From these graphs, it is possible to verify that the model had learned about the training data, allowing it to make predictions of TEC data.

As mentioned earlier, to evaluate the effectiveness of the proposed MLP-NN model in the prediction of VTEC values, the metric adopted was the MAE. The VTEC values from the maps described in Section 2 were used as the reference values ($\vartheta_j$), while the predicted samples ($\varsigma_j$) came from the MLP-NN, NeQuick G model, and GIM. The errors computed consider the values from the models in the spatial grid covering from 33°W to 70°W longitude and from 4°N to 32°S latitude with 1° resolution.

## 4. Model Evaluation

Here we present the VTEC predictions obtained based on the MLP-NN described in the previous section. The performance of the proposed model will be evaluated according to the months of the year, location, and as function of the time of day (dawn, morning, afternoon, and night). In addition, the performance of the NeQuick G model and GIM are also presented for comparison purposes. Please remember that the VTEC maps built with real TEC values were used as the reference for error calculation.

### 4.1. Evaluation of Ability of Seasonal Representation

Figure 4 presents histograms with errors for the MLP-NN for all forecasted maps for the whole year of 2014. The figure is divided into months (panels) and hours (color-coded curves). The VTEC errors (VTEC map–VTEC predicted) were binned in 4 intervals, representative of substantial TEC changes over low-latitude regions, namely, dawn (03:00–08:59 UT), morning (09:00–14:59 UT), afternoon (15:00–20:59 UT), and night (21:00–02:59 UT). Please note that the local time may be between UT-4:40 and UT-2:12, depending on the longitude considered.

The results indicate that between May and August (winter solstice months) the errors are smaller (concentrated around zero) when compared to the remaining months (equinox and summer solstice); this is especially true for nighttime. This behavior is probably related to the fact that during the winter solstice months the occurrence of highly variable phenomena such as spread F and plasma bubbles are rare [30]. As an example, during June nights, approximately 82% of the model error is less than 5 TECu for the afternoon; therefore, the results agrees very well with real data. In contrast, during months along the equinox and summer solstice, only 59% of the model errors are less than 5 TECu for the afternoon.

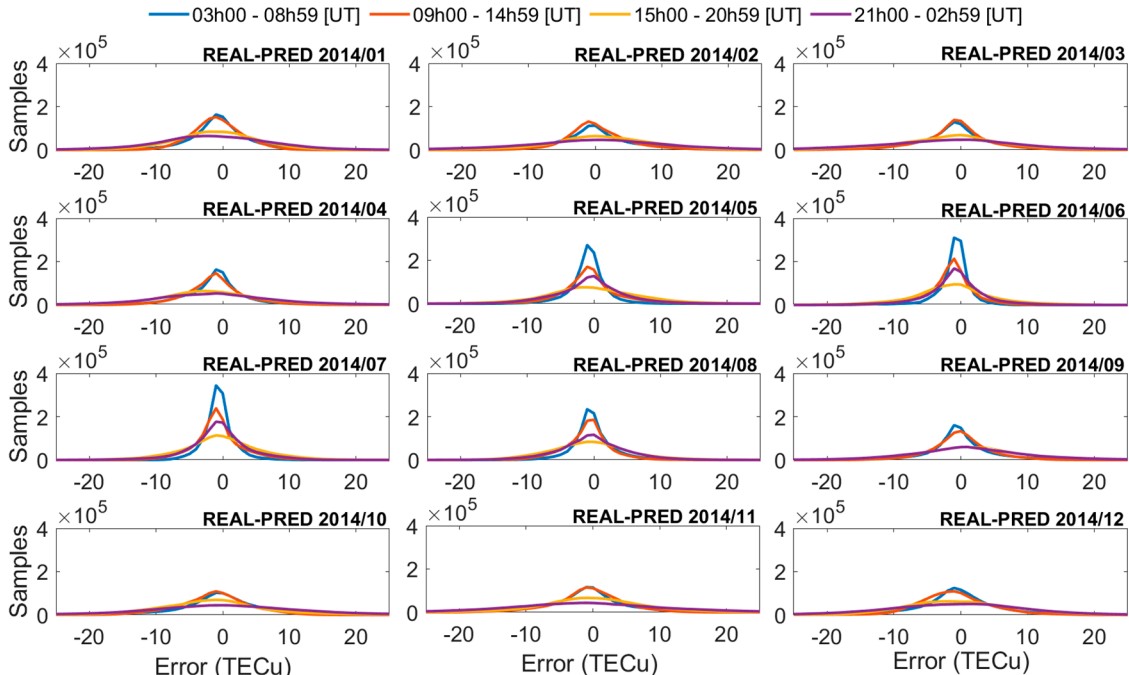

**Figure 4.** TEC errors according to month (panels) and time interval (colors) for the year 2014. The MLP-NN-based model presents good agreement with the data, especially during the winter solstice, as expected.

Continuing with the model evaluation, Figure 5 shows the average MAE ($\overline{\text{MAE}}$) value for the proposed MLP-NN (blue bars), as well as for the NeQuick G (red bars) and GIM (yellow bars). In this figure, the $\overline{\text{MAE}}$ values were calculated considering the entire coverage of the TEC maps generated by the MLP-NN, GIM or NeQuick G, i.e., the

$\overline{\text{MAE}}$ values correspond to the mean errors of the entire representation for distinct time intervals and months, revealing the seasonal and temporal average distribution of the errors. It is readily discernible that the $\overline{\text{MAE}}$ values for the predictions by the MLP-NN are considerably smaller than the other models for all months and for all time intervals, i.e., the MLP-NN results are consistently better. The improved prediction by the MLP-NN is especially noticeable for the months between October and March. As an example, the MLP-NN $\overline{\text{MAE}}$ values are up to 76% lower than those from the GIM. Indeed, for any time interval, the MLP-NN $\overline{\text{MAE}}$ values remain stable throughout the year. This stability is not observed for the $\overline{\text{MAE}}$ values obtained using the NeQuick G and GIM. Please note that the $\overline{\text{MAE}}$ values from these models have a substantial increase in the spread F season months.

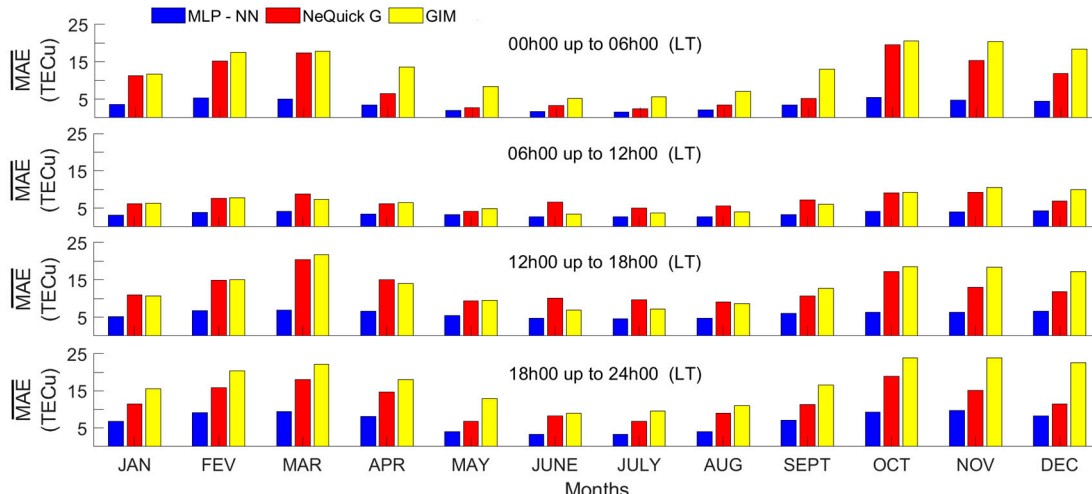

**Figure 5.** Average MAE ($\overline{\text{MAE}}$) from the MLP-NN, NeQuick G, and GIM for the year 2014 considering the entire coverage over Brazil. The panels are organized in 4 time intervals to illustrate the seasonal and temporal improvements obtained with the usage of the MLP-NN.

### 4.2. Evaluation of the Error from a Spatial Coverage Perspective

In the previous section, MAE calculation was performed considering the entire coverage as a unique set, i.e., only one value was obtained for each time available. Notwithstanding, the GPS errors introduced by the ionosphere are dependent on the TEC distribution, and the ionosphere over low-latitude regions is known to be highly variable [11]; therefore, the VTEC spatial distribution must be evaluated.

To evaluate the efficiency of the MLP-NN model in producing an accurate spatial representation of the TEC, the MAE values in this section were calculated for each point in the grid individually so that the spatial distribution of the model errors could be obtained. For this analysis, the MAE for each day was calculated over the entire spatial grid (at each latitude and longitude with 1° resolution).

Figure 6 shows examples of spatial errors (MAE) from the MLP-NN (right panels), NeQuick G (middle panels), and GIM (left panels) for two distinct days in 2014 (chosen arbitrarily). The days exhibited in Figure 6 are, respectively, 1 June (upper panels) and 30 November (lower panels), 2014. These days correspond to two distinct seasonal ionospheric conditions; the first represents a winter solstice condition while the second corresponds to a summer solstice. The spatial MAE values confirm the results of Figure 5, showing that the GIM and NeQuick G present inferior performances when compared to the MLP-NN. The MLP-NN MAE values were considerably lower for both days analyzed in this example, corresponding to better accuracy for both distinct seasonal conditions. Especially during larger TEC conditions, as represented in the lower panels (summer solstice), the MLP-NN results are better in the entire spatial grid. It must be mentioned that this seasonal period is contained in the spread F season, when the ionosphere causes problems for GPS transmissions more often [31].

During winter solstice, notice that the errors from the GIM (upper left panel) are mostly concentrated in the regions close to the magnetic equator, while those from NeQuick G (upper middle panel) are condensed in the EIA region. For the period of summer solstice, the errors exhibit the opposite trend, i.e., the GIM (lower left panel) and NeQuick G (lower middle panel) errors are more concentrated in the EIA and magnetic equator regions, respectively. The errors from MLP-NN, besides considerably lower, are more concentrated at low latitudes for both seasonal conditions. This is also more coherent, because these regions present larger TEC values consistently, along both seasons represented.

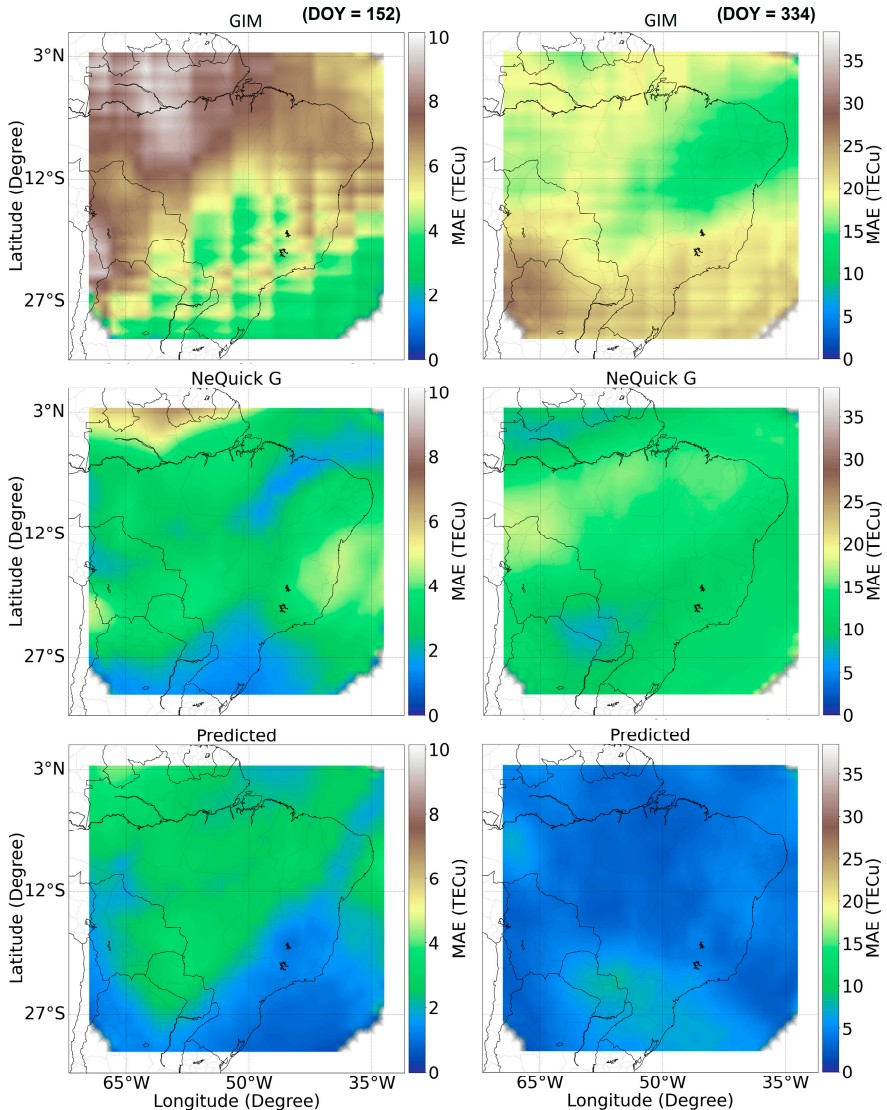

**Figure 6.** MAE for each point of the spatial grid coverage for the DoYs 152 (**right panels**) and 334 (**left panels**) as obtained by the MLP-NN (**lower panels**), NeQuick G (**middle panels**) and GIM (**upper panels**) for the year 2014 over Brazil.

### 4.3. Evaluation of the Error According to F10.7 and Ap Indexes

The ionospheric TEC distribution is also dependent on solar flux and geomagnetic activity [32]; hence, this section investigates the model representation performance regarding these aspects, as the F10.7 and Ap indexes are also used to feed the MLP-NN.

Table 2 presents the $\overline{\text{MAE}}$ together with the standard deviation ($\sigma$) for the models analyzed in terms of the F10.7 index. The MLP-NN improvement percentage was calculated following $100 \times \left(\overline{\text{MAE}}_{\text{M}} - \overline{\text{MAE}}_{\text{MML-NN}}\right) / \overline{\text{MAE}}_{\text{M}}$. Notice that the MLP-NN model had $\overline{\text{MAE}}$ reductions over 50% for all the conditions of F10.7. Supporting the improved accuracy

of the MLP-NN model, the standard deviation results presented similar behavior, with much smaller values for this model when compared to the others.

**Table 2.** Evaluation of the accuracy of the methods for different ranges of solar flux.

| F10.7 (s.f.u.) | Data Source | $\overline{MAE}$ (TECu) | σ[MAE] (TECu) | MLP-NN Improvement (%) |
|---|---|---|---|---|
| F10.7 ≥ 150 | MLP-NN | 5.44 | 3.39 | - |
|  | NeQuick G | 11.44 | 6.25 | 52.44 |
|  | GIM | 14.77 | 8.29 | 63.17 |
| 100 ≤ F10.7 < 150 | MLP-NN | 4.68 | 3.00 | - |
|  | NeQuick G | 9.72 | 6.28 | 51.85 |
|  | GIM | 11.35 | 7.12 | 58.77 |
| F10.7 < 100 | MLP-NN | 2.22 | 1.53 | - |
|  | NeQuick G | 6.48 | 5.19 | 65.74 |
|  | GIM | 4.94 | 2.42 | 55.06 |

Regarding the geomagnetic condition, the data were grouped in a first set containing only the days under "quiet" geomagnetic conditions (Ap < 27) and in a second set with only the days under "disturbed" geomagnetic conditions (Ap ≥ 27) [33].

Table 3 presents the $\overline{MAE}$, together with the standard deviation (σ), for the models analyzed in terms of the Ap index. Under geomagnetic disturbed conditions, the VTEC prediction was admittedly more challenging. The observed MLP-NN $\overline{MAE}$ for this condition revealed improvements of 45.38% and 38.80% when compared to the values for the GIM and NeQuick G, respectively. For geomagnetic quiet conditions, the improvement provided by the MLP-NN $\overline{MAE}$ reached 60.99% and 52.52% when compared to the values for the GIM and NeQuick G, respectively. For both geomagnetic conditions, note again that the MLP-NN model had much smaller errors than the other models analyzed. The standard deviation for the MLP-NN model was also smaller than those from the NeQuick G and from GIM. These results indicate that the MLP-NN model is also better in representing the ionospheric TEC under distinct solar and geomagnetic conditions.

**Table 3.** $\overline{MAE}$ values for different TEC methods considering geomagnetic quiet and disturbed cases.

| Ap Index | Data Source | $\overline{MAE}$ (TECu) | σ[MAE] (TECu) | MLP-NN Improvement (%) |
|---|---|---|---|---|
| Ap ≥ 27 | MLP-NN | 8.28 | 3.83 | - |
|  | NeQuick G | 13.53 | 4.80 | 38.80 |
|  | GIM | 15.16 | 6.57 | 45.38 |
| Ap < 27 | MLP-NN | 4.90 | 3.18 | - |
|  | NeQuick G | 10.32 | 6.33 | 52.52 |
|  | GIM | 12.56 | 7.84 | 60.99 |

## 5. Discussion

In the previous sections, the results indicate that the MLP-NN model may successfully represent the VTEC over Brazil with better accuracy than models and maps widely used such as GIM and NeQuick G. In this discussion section, some applications of the MLP-NN model are addressed. The first step in this discussion is to compare the predictions of this model with the reference VTEC maps built according to [20]. The purpose of this

evaluation is to show, explicitly, the ability of the MLP-NN to properly represent the spatial distribution of the TEC over the Brazilian region.

Figures 7 and 8 illustrate two days (arbitrarily chosen) corresponding to periods of winter solstice (June 6) and summer solstice (November 30) in the year 2014.

The left panels in Figure 7 exhibit the VTEC maps estimated with real data using the methodology presented in [20]. From upper to lower panels, the periods represented are post-midnight time (7h00 UT), morning (13h00 UT), afternoon (20h00UT), and early nighttime (23h00UT), respectively. Please remember that the local time is between UT-4:40 and UT-2:12. The color bar at right describes the VTEC values. The right panels use the same graphical elements, however, this time presenting the prediction obtained with the MLP-NN model. It is evident that all features are properly preserved and that the spatial structure of the TEC predicted with the MLP-NN is coherent with the physical processes over low-latitude regions.

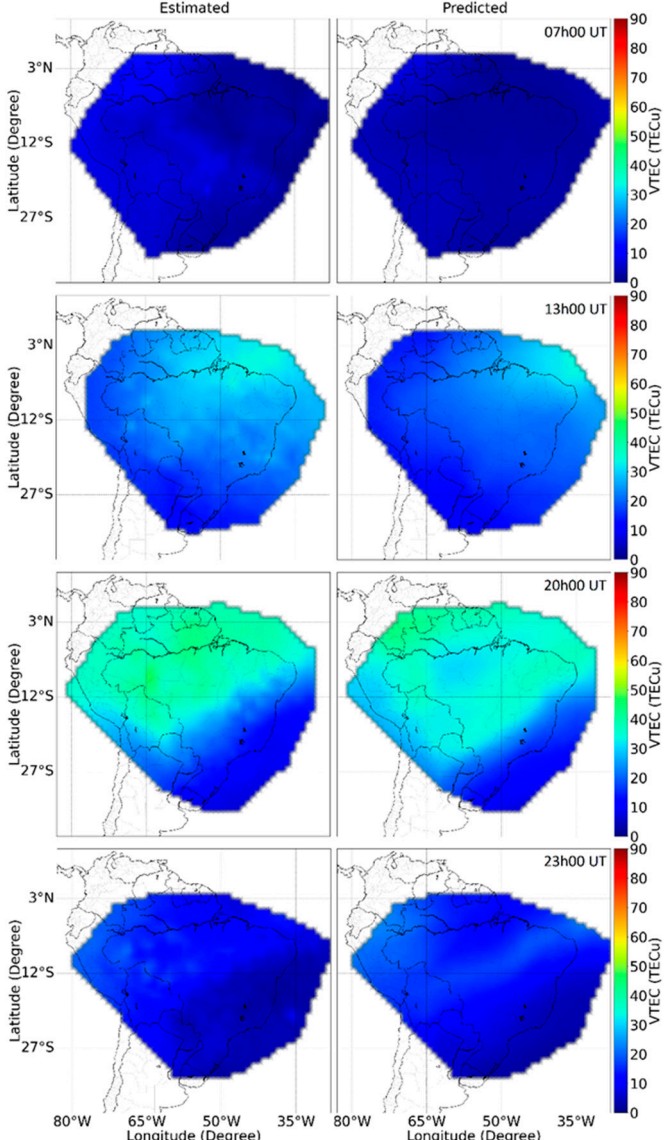

**Figure 7.** Comparison between the real VTEC maps (**left**) and the VTEC maps predicted with the MLP-NN (**right**) during a winter solstice arbitrary day (6 June 2014). From upper to lower panels, frames corresponding to the post-midnight, morning, afternoon, and early nighttime are presented.

Figure 8 is similar to the previous one, except that the day represented belongs to a summer solstice (30 November 2014). The graphical representation is the same as that used in the previous figure.

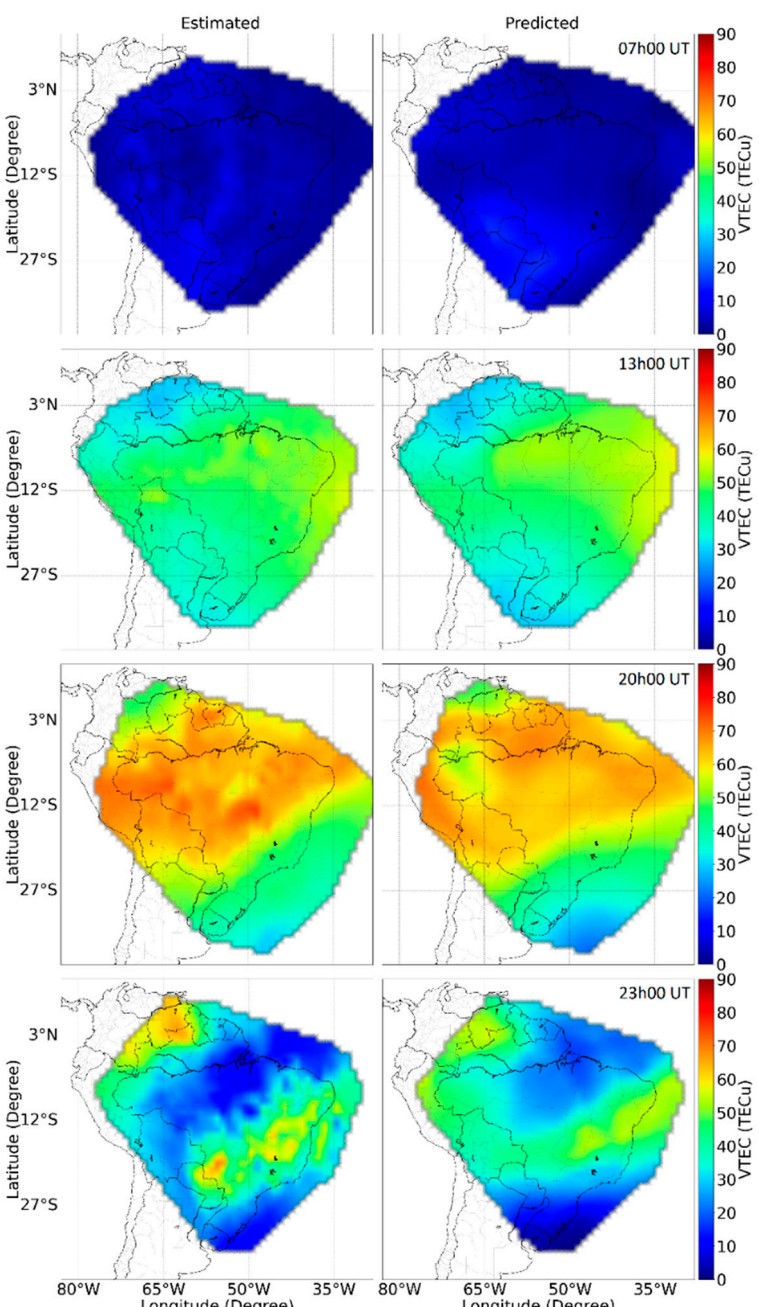

**Figure 8.** Comparison between the real VTEC maps (**left**) and the VTEC maps predicted with the MLP-NN (**right**) during a summer solstice arbitrary day (30 November 2014). From upper to lower panels, frames corresponding to the post-midnight, morning, afternoon, and early nighttime are presented.

Please observe that aspects such as the concentration of the ionospheric plasma over regions around the geomagnetic equator and its posterior redistribution by the fountain effect to low latitudes, away from the equator, are evident in the results provided by the MLP-NN model (right panels). Hence, the ability of the MLP-NN to reproduce the TEC spatial structure is confirmed, regardless of the seasonal condition of the ionosphere.

These results are important and suggest that, although the ionospheric phenomenology is complex and widely variable over the Brazilian region, the configuration of the MLP-NN is adequate to reproduce its effects. It must be mentioned, however, that this ability is a heritage from the dataset feeding the network and the ionospheric behavior was learned implicitly.

*Positioning Performance*

One of the main interests in the development of this type of prediction model is the use for ionospheric correction in satellite navigation algorithms. To confirm the better accuracy of results from the MLP-NN model, the values of this prediction were tested in two locations in Brazil, Recife (8.05°S, 34.95°W) and Salvador (12.97°S, 38.51°W). These stations were used because data were available for a long period with good quality records. The positioning results from the MLP-NN model and from NeQuick G and GIM were compared, because the last two are widely used in positioning applications. The positioning method used in this evaluation was the GPS single point positioning (SPP) [34]. In this approach, observables from the L1 GPS signal code were used together with the transmitted ephemeris to compute the position. In this processing, the ionosphere corrections were made considering the MLP-NN, NeQuick G and GIM independently, while the tropospheric corrections used the Hopfield model [35]. In addition, the VTEC values estimated by the reference VTEC map constructed as described in Section 2 were also used and will be referenced in this section as TEC MAP.

Figure 9 shows an example of the SPP 3D error for 20 November 2014, on a receiver located in Recife. Here, 3D error refers to the value considering $\sqrt[2]{(e^2 + n^2 + u^2)}$ for the east (e), north (n) and up (u) directions, respectively. This example illustrates how the ionosphere influences the positioning error. During the day, from 05 UT to 15 UT, the errors were smaller from all the models. This behavior agrees with the MAE graphs in Figure 6, which show that approximately during these hours the MLP-NN model presented smaller mean absolute error values. However, as the ionosphere becomes more ionized during the afternoon (15UT to 20 UT), as shown in the central panels of Figures 7 and 8, the positioning errors increased. The 3D errors in the dusk and early nighttime periods increased probably due to enhancement in the ionospheric plasma motion at these hours. This motion is augmented by the vertical component of the plasma drift during the prereversal enhancement of the zonal electric field [36]. Please observe that this vertical drift increase is highly variable and promotes a large redistribution of plasma to latitudes away from the equator; therefore, any model produces slightly poor representation at these hours. Subsequent to the prereversal enhancement, the equatorial and low-latitude ionosphere often experiences instabilities that lead to the formation of EPBs. EPBs are huge regions of depleted plasma and are extremely difficult to predict; hence, again, any model is not capable of presenting a perfect representation due to these drastic events that change the plasma spatiotemporal structure. For these reasons, errors up to ~3UT are large for all the models considered.

The smallest positioning errors were typically observed for the TEC MAP (blue line) followed by the MLP-NN (yellow line) that was trained using TEC MAP data as inputs. The TEC MAP presented the best positioning result with an average error of 3.37 m; however, because processing is required, it cannot broadcast instantaneous positioning information. Among the analyzed models, the MLP-NN outperformed the NeQuick G (purple line) and GIM (orange line) with an average error of 4.10 m while the others presented errors of 5.59 m and 6.10 m, respectively. Therefore, the MLP-NN model result presented an error in positioning that was 27% and 33% better (smaller) when compared to NeQuick G and GIM, respectively.

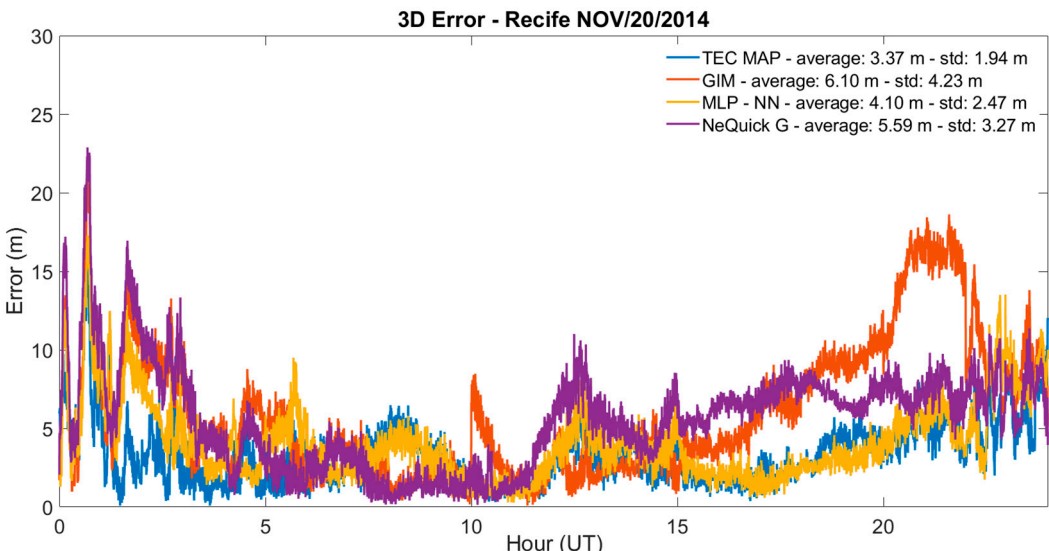

**Figure 9.** Example of 3D single point positioning error for 20 November 2014, over the RBMC station in Recife. The error of the proposed model (yellow line) is typically smaller than the NeQuick G (purple line) and GIM (orange line).

Following the analysis of the SPP, Figures 10 and 11 show the cumulative distribution function (CDF) for two days (arbitrarily chosen), representing winter (upper panels) and summer (lower panels) solstices, respectively. The CDFs of the 3D positioning errors were evaluated for the stations in Recife (Figure 10) and Salvador (Figure 11). The results were in general agreement with those from Figure 9, i.e., the TEC MAP and the MPL-NN showed the best positioning with smallest errors when compared to the other models. In the upper panels showing the results for June 6 (winter solstice), the errors were small for all the models when compared to the errors for the summer solstice condition. This result was expected; please observe that a comparison between the maps in Figures 7 and 8 reveals that the VTEC variability in the winter season is much smaller than during the summer. Regarding the TEC estimator performance, considering, for example, the summer solstice condition (12 November 2014) in Salvador, the TEC MAP and the MPL-NN had probabilities of 3D errors < 5 m reaching 83% and 79%, respectively. These values were considerably larger than the values of 67% and 72% obtained with the GIM and the NeQuick G, respectively, i.e., errors smaller than 5 m were substantially more expected for the MLP-NN model. The comparison shows that SPP users would have improvements reaching 12 and 7 percentage points, respectively, using the MPL-NN for positioning at low latitudes.

A different analysis of the positioning performance is presented in Figures 12 and 13 showing the average value of 3D positioning errors for two full months according to the hour of the day. The stations in Recife (Figure 12) and Salvador (Figure 13) and the months of June (left panels) and November (right panels) were used again to represent periods around winter and summer solstices, respectively. Notice that the MLP-NN 3D average error (yellow line) is typically smaller than that of NeQuick G (purple line) or GIM (orange line). This is especially noticeable in the periods where the VTEC is typically enhanced due to photoionization. During these hours the MLP-NN performance was the best for the positioning when compared to the other data sources. Consider, for example, the month of November at 18UT (Figure 13, lower panel), the average 3D error was 3.14 m for the MLP-NN, while the errors for the GIM and NeQuick G were, respectively, 6.08 m and 6.35 m. Therefore, the error of the MLP-NN model was near the half that of the other evaluated methods.

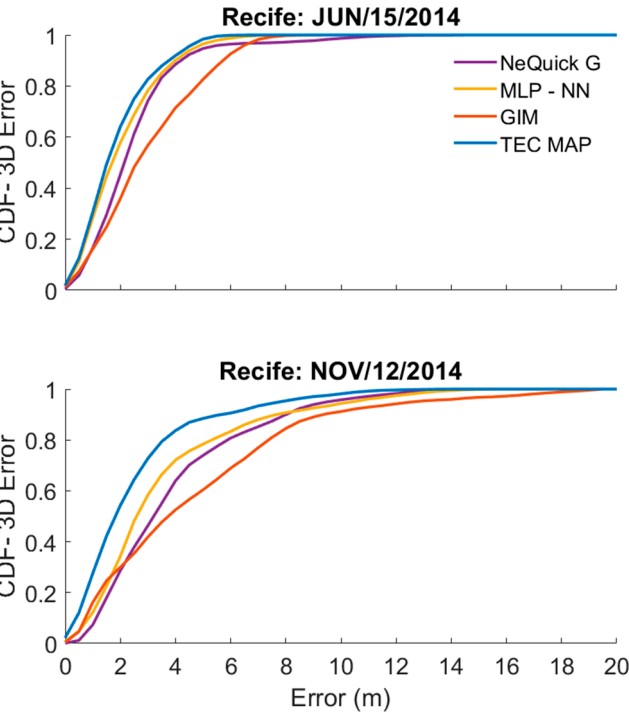

**Figure 10.** Empirical CDF of the SPP 3D error considering different ionospheric models for the station in Recife for the winter (15 June at upper panel) and summer (12 November at bottom panel) solstice conditions in 2014.

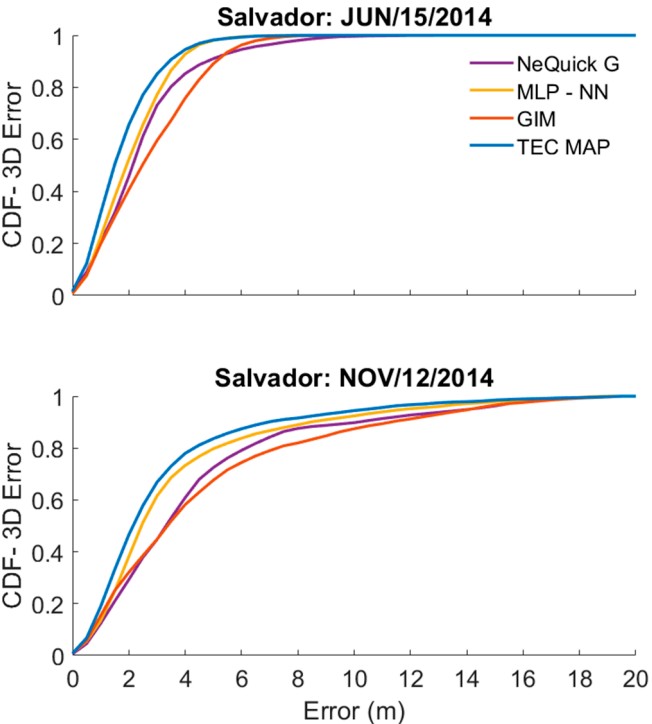

**Figure 11.** Empirical CDF of the SPP 3D error considering different ionospheric models for the station in Salvador for the winter (15 June at upper panels) and summer (12 November at bottom panel) solstice conditions in 2014.

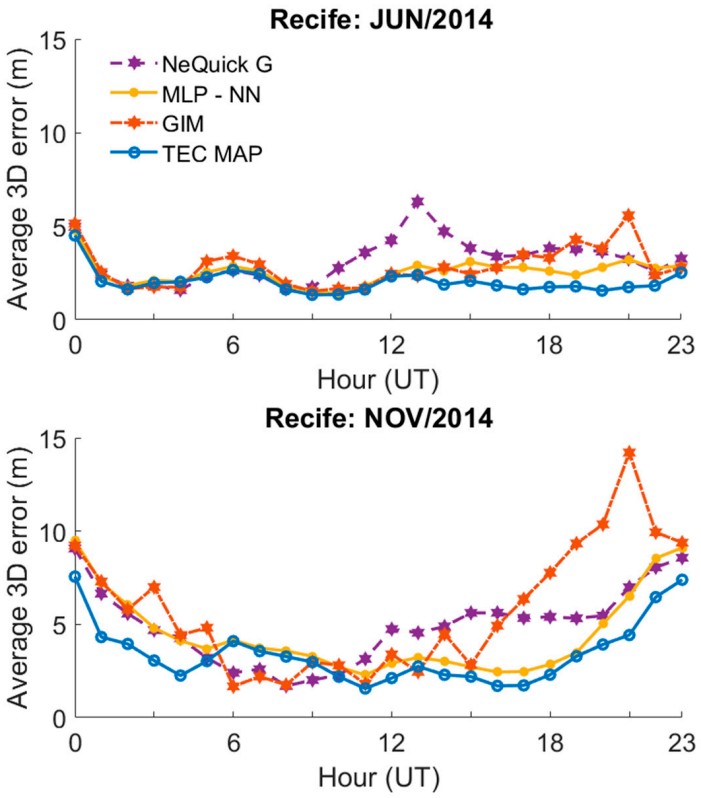

**Figure 12.** Average 3D error as function of the hour for the station in Recife (**bottom panels**) during the months of June (**upper panel**) and November (**bottom panel**) for the year 2014.

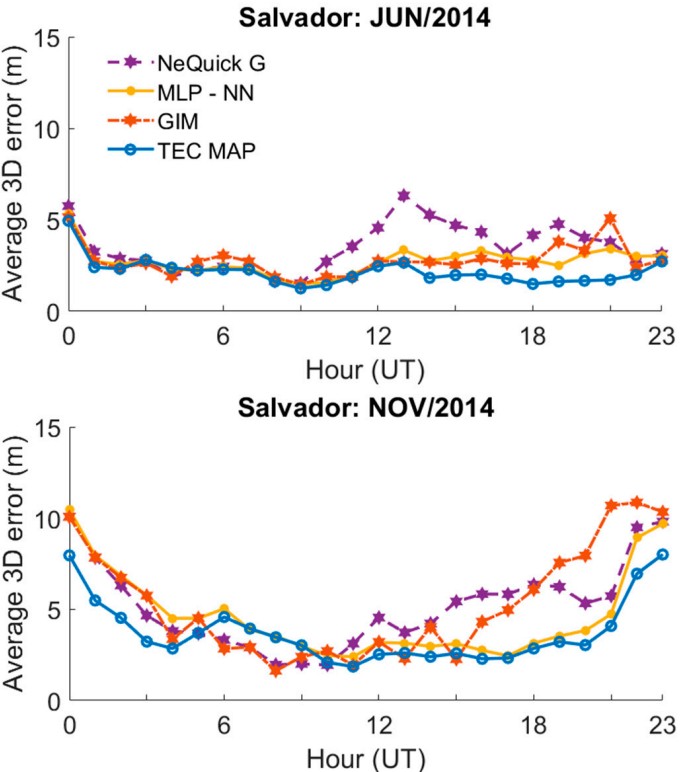

**Figure 13.** Average 3D error as function of the hour for the stations in Salvador during the months of June (**upper panel**) and November (**bottom panel**) for the year 2014.

Table 4 shows the average and standard deviation of the 3D errors for both stations (Recife and Salvador) for the months of June and November 2014. These results indicate that the SPP achieves better accuracy and precision using the MLP-NN when compared to the other TEC estimates.

**Table 4.** Average and standard deviation for 3D error during the months of June and November in Recife and Salvador during 2014.

| 3D Error | MODEL | Recife | | Salvador | |
|---|---|---|---|---|---|
| | | MEAN (m) | STD (m) | MEAN (m) | STD (m) |
| JUN | GIM | 2.81 | 1.94 | 2.77 | 2.03 |
| | NeQuick G | 3.16 | 2.26 | 3.50 | 2.55 |
| | MLP-NN | 2.53 | 1.82 | 2.72 | 2.04 |
| | TEC MAP | 2.03 | 1.62 | 2.15 | 1.86 |
| NOV | GIM | 5.71 | 3.95 | 5.30 | 3.84 |
| | NeQuick G | 4.92 | 3.08 | 5.17 | 3.45 |
| | MLP-NN | 4.47 | 3.26 | 4.65 | 3.47 |
| | TEC MAP | 3.43 | 2.51 | 3.73 | 2.74 |

Finally, it must be mentioned that this approach was designed to provide services, such as files equivalent to those from GIM, and might be useful to users of positioning systems over the Brazilian region.

## 6. Concluding Remarks

The equatorial and low-latitude ionosphere over the Brazilian region presents very particular electrodynamics involving several mechanisms. All this variability is difficult to be properly addressed by models available nowadays; therefore, the information about the spatiotemporal structure of the TEC has been poorly covered so far. Recent works applied neural network and deep learning techniques to improve TEC estimation considering some stations over the Brazilian region during some specific periods of the year (e.g., [37,38]). In this work an alternative neural network approach was proposed to provide better forecasting of vertical TEC values over the entire Brazilian region, considering all the year of 2014 (solar maximum period), for both quiet and disturbed geomagnetic periods, under distinct solar flux conditions. In addition, the estimation of the improvement in single point positioning was also discussed. The proposed method to obtain the vertical TEC values was the use of a multilayer perceptron neural network (MLP-NN) fed with TEC MAPs (Marini-Pereira et al.) from the five days prior to the day to be predicted and with the following parameters of space weather: day of the year, hour, solar F10.7 index and the geomagnetic index Ap. The vertical TEC predictions were made for the entire year of 2014 across the entire territory of Brazil with 1° resolution. The performance of MLP-NN was compared to the results obtained from NeQuick G and from GIMs in the IONEX format. The metric used for this comparison was the mean absolute error (MAE).

The results may be summarized as follows:

(1) The average monthly MAE values are smaller (i.e., better) for the proposed MLP-NN for all the months, in all time intervals considered when compared to NeQuick G and to GIM. In some cases, the MLP-NN MAE was 76% less than the GIM MAE.

(2) The analyses of MAE over the entire Brazilian region (e.g., Figure 6) considering two entire days during distinct seasons (June and November) reveals that the MLP-NN spatial error is also qualitatively better than the NeQuick G and GIM.

(3) The evaluation considering distinct solar flux levels reveals MAE values 51.85% better than those from the other data sources (NeQuick G and GIM).

(4)  The analysis considering distinct geomagnetic index levels indicate MAE values that are 38.80% and 52.52% better than the other data sources for disturbed and quiet geomagnetic periods, respectively.

(5)  The analyses of the 3D SPP error are a new feature presented in this work and the results indicate that positioning errors using the vertical TEC forecasted by the proposed MLP-NN are remarkably similar to those obtained using real data of the TEC MAPs. Please observe that the MLP-NN is providing the values in advance (one day ahead).

(6)  Considering the example case of November 20, 2014, the single point positioning analysis showed that 3D SPP errors from the MLP-NN were 27% and 33% better (smaller) when compared to NeQuick G and GIM, respectively. For the monthly evaluation improvements up to 22% (June) and 21% (November) were achieved on the same basis of comparison.

(7)  The predicted vertical TEC maps using the MLP-NN reproduce the spatiotemporal TEC structure expected over this region, which is not obtained using the other models (e.g., [11]).

This work demonstrates the feasibility of the use of deep learning methods to forecast improved vertical TEC maps to be used in positioning approaches over the Brazilian region. The results are promising even considering distinct solar flux, geomagnetic conditions, and season of the year.

**Author Contributions:** Conceptualization, A.S., A.M. and B.V.; methodology, A.S., A.M.; software, A.S. and C.F.J.; validation, A.S., A.M. and J.S.; formal analysis, A.S., A.M., J.S., M.M., B.V. and C.F.J.; investigation, A.S., A.M., J.S., M.M., B.V. and C.F.J.; data curation, A.S.; writing—original draft preparation, A.S., A.M. and J.S.; writing—review and editing, A.S., A.M., J.S., M.M., B.V. and C.F.J. All authors have read and agreed to the published version of the manuscript.

**Funding:** This research received no external funding.

**Data Availability Statement:** Data from RBMC can be accessed at https://www.ibge.gov.br/geociencias/informacoes-sobre-posicionamento-geodesico/rede-geodesica/16258-rede-brasileira-de-monitoramento-continuo-dos-sistemas-gnss-rbmc.html?=&t=downloads (accessed on 14 December 2022).

**Acknowledgments:** A. L. A. Silva is grateful to CNPq scholarship (141442/2020-4 and 132896/2018-4) and worked on this research in collaboration to the framework CNPq 465648/2014-2 and FAPESP 2017/01150-0. A. O. Moraes is supported by CNPq award 309389/2021-6 and C. Faria Jr 102204/2021-7 and 153037/2021-0. J. Sousasantos acknowledges FAPESP 2018/06158-9 and NSF (AGS-1916055). The authors acknowledge the Instituto Brasileiro de Geografia e Estatística (IBGE) for providing GNSS data from the RBMC network, used for TEC estimation. IGS data. We acknowledge the INCT (In-stituto Nacional de Ciência e Tecnologia) for the support for the development of the research.

**Conflicts of Interest:** The authors declare no conflict of interest.

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
