# Peer review of "Using Deep Learning to Map Ionospheric Total Electron Content over Brazil"

_remotesensing, doi:10.3390/rs15020412_

Round 1

Reviewer 1 Report

Comments to the manuscript remotesensing 2130824

Using deep learning to map ionospheric Total Electron Content over Brazil

by Andre Silva, Alison Moraes, Jonas Sousasantos, Marcos Maximo, Bruno Vani, Clodoaldo Faria Jr.

Deep learning is a powerful tool for predicting the behavior of parameters in a variety of science and engineering fields and provides a very large set of methods and architectures within each method. The parameter the total electron content (TEC) provides time series, which is a good testing ground for investigating the effectiveness of the methods. This paper focuses on the development of a method for predicting TEC maps over Brazil, an area with specific features of ionospheric behavior. Typically, authors appeal to the need to increase positioning accuracy to motivate publication, but do not make appropriate evaluations. This paper makes such estimates, and in comparison with the widely used IONEX and NeQuick G models.

The paper is recommended for publication, although not without some comments.

Comments

1.Line 190, Section 2: the metrics themselves are not specified.

2.Line 263: does this refer to normalized values?

3.Line 277: replace 3.1 with 3.3.

4. Line 349: the comparison would be clearer if at least within one row the range of the vertical axes was the same.

5. Line 447: how are your maps different from the maps presented in [12] Marini-Pereira et al. (2020)?

Some inaccuracies

1. Line 13: replace chal-lenge with challenge.

2. Line 21: replace season-ality with seasonality.

3. Line 155: replace it with its.

4. Line 290: separate replace with separated.

5. Line 310: Pease replace with Please.

Reviewer 2 Report

The authors employ temporal series of TEC to build an independent TEC estimator model for low latitudes using a deep learning framework. The work done in the manuscript looks good and may qualify for acceptance for publication. The authors are recommended to incorporate the following comments in their subsequent submission.

 Line 11  The statement looks odd “The spatiotemporal distribution of TEC has a strong influence over the solar activity and the geomagnetic field.”

Line 13  chal-lenge =>challenge

Line 21  season-ality => seasonality

Lines 24-25 Rephrase the sentence

Line 28 The authors describe in abstract about the single point positioning but the key word for the same is missing

Line 44-51 I suggest the authors to support the statement with an appropriate citation. The authors may use Vankadara et al, (2022) https://doi.org/10.3390/rs14030652  that discusses most of the phenomena listed in this sentence.

Line 55 IONEX is not a model, its simply an exchange format. The authors should write it GIMs. There are a wide range of mathematical, physics based, and data driven empirical models available at present which has been investigated/implemented by researchers around the globe (IRI, IRI-Plas, NTCM, GITM, SVM, LSTM,  etc. which the authors need to reiterate to have a wide visibility of the audience. To assist the authors, I suggest to have a look at the paper by Reddybattula et al., 2022 (https://doi.org/10.3390/universe8110562) and Dabbakuti et al. 2021 (https://doi.org/10.1016/j.actaastro.2020.08.034) to summarize a range of models and cite them appropriately.

Line 55a The authors are expected to present the plasmaspheric contribution to TEC which is not explained by many of the models developed from ground-based datasets. There is indeed a visible contribution from plasmasphere. For, further information, the authors may refer Belehaki et al. 2004 (https://doi.org/10.1016/j.asr.2003.07.008) and place appropriately in their introduction.

Line 56 mixing up with other models “provided by differente centers with a given sample resolution, such as of about 2 hours.”. Moreover, spelling of different is incorrect.

Line 63 Incorrect citation style M. R. Ghaffari Razin and B. Voosoghi [5]

Line 65 The authors mention it as AUTHOR ET AL. [6] in cases where the citation comes at the beginning or middle of the sentence.

Why did the authors consider 2-hourly GIMs where 15min GIMs are available now a days.

Line 489 Do the authors used their own program to calculate the single frequency receiver positioning, I did not find anything about the same in methodology. If the authors used any online resources or program, please mention it.

What is the uncertainty level of TEC values from the map over a station whose data is used for VTEC calculations. This will guide me on the accuracy of the method followed for developing the regional TEC maps.

Figure 9 I expect the error plots for specimen dates of different seasons.

The conclusion section looks weak, I suggest improving it as much as possible.
